# Progress of Wnt Signaling Pathway in Osteoporosis

**DOI:** 10.3390/biom13030483

**Published:** 2023-03-06

**Authors:** Yongguang Gao, Na Chen, Zhanda Fu, Qing Zhang

**Affiliations:** Tangshan Key Laboratory of Green Speciality Chemicals, Department of Chemistry, Tangshan Normal University, Tangshan 063000, China

**Keywords:** osteoporosis, Wnt signaling pathway, osteoblasts, osteoclasts, Dkk-1, SOST

## Abstract

Osteoporosis, one of the serious health diseases, involves bone mass loss, bone density diminishing, and degeneration of bone microstructure, which is accompanied by a tendency toward bone fragility and a predisposition to fracture. More than 200 million people worldwide suffer from osteoporosis, and the cost of treating osteoporotic fractures is expected to reach at least $25 billion by 2025. The generation and development of osteoporosis are regulated by genetic factors and regulatory factors such as TGF-*β*, BMP, and FGF through multiple pathways, including the Wnt signaling pathway, the Notch signaling pathway, and the MAPK signaling pathway. Among them, the Wnt signaling pathway is one of the most important pathways. It is not only involved in bone development and metabolism but also in the differentiation and proliferation of chondrocytes, mesenchymal stem cells, osteoclasts, and osteoblasts. Dkk-1 and SOST are Wnt inhibitory proteins that can inhibit the activation of the canonical Wnt signaling pathway and block the proliferation and differentiation of osteoblasts. Therefore, they may serve as potential targets for the treatment of osteoporosis. In this review, we analyzed the mechanisms of Wnt proteins, *β*-catenin, and signaling molecules in the process of signal transduction and summarized the relationship between the Wnt signaling pathway and bone-related cells. We hope to attract attention to the role of the Wnt signaling pathway in osteoporosis and offer new perspectives and approaches to making a diagnosis and giving treatment for osteoporosis.

## 1. Introduction

Osteoporosis is a serious, healthy disease that involves low bone mass and microstructure degeneration of bone tissue and is accompanied by a predisposition to fracture [1]. Fractures and bone defects induced by osteoporosis have gradually become one of the most widespread medical problems due to their poor histological basis and decreased osteoblast activity. Overall, about one-third of people aged 60 to 70 years are suffering from osteoporosis, and more than half of the elderly over 80 years old are suffering from osteoporosis [2]. With the extension of human life and the worsening of an aging society, the number of people suffering from osteoporosis is increasing. It is predicted that by 2050, the number of people suffering from osteoporotic fractures will increase from 60 million to more than 120 million [3], and the cost of treating osteoporotic fractures is expected to reach at least $25 billion by 2025 [4]. This seriously affects people’s living standards and quality of life. Therefore, we need to explore and clarify the pathogenesis of osteoporosis and formulate effective treatment strategies.

The pathogenesis of osteoporosis is caused by osteoblast apoptosis and osteoclast proliferation in the body, which leads to bone metabolism disorder and bone loss, resulting in the deterioration of bone morphology [5]. During life, from birth through growth, development, and maturity to aging, the bone is constantly repeating the process of bone remodeling. This process mainly consists of four phases [6]: as shown in Figure 1, the bone resorption phase, the reversal phase, the matrix deposition phase, and the bone mineralization phase. The bone-related cells in bone tissue participate in bone remodeling through complex interactions [7] and can transform each other under certain conditions. During the process of bone remodeling, the number and function of the osteoblasts and the osteoclasts maintain a dynamic balance, which is very important for the microenvironment of the bone matrix. If the microenvironment in the bone matrix is abnormal, bone-related diseases such as osteoporosis will occur.

## 2. Bone Growth Factors

The formation and development of osteoporosis are regulated by various bone-related transcription factors and genetic factors at the molecular level, including transforming growth factor *β* (TGF-*β*), parathyroid hormone (PTH), and fibroblast growth factor (FGF) [8,9,10]. These bone-related growth factors have direct or indirect physiological and pathological regulatory effects on osteoblast differentiation and the activities of osteoblasts and osteoclasts through autocrine/paracrine mechanisms [11]. When the normal regulatory function is blocked, it will lead to an imbalance of bone formation and bone absorption and eventually induce osteoporosis.

### 2.1. Transforming Growth Factor β (TGF-β)

TGF-*β* is widely found in human normal tissues and cells, especially in bone tissues and platelets [12]. There are more than 40 members in the TGF-*β* superfamily: TGF-*β*, activin, inhibin, bone morphogenetic protein (BMP), and so on. These family numbers are polypeptide compounds with similar structures and different functions. The newly secreted TGF-*β* gene (pre-pro-TGF-*β*) combines with the latency-associated protein (LAP) and forms a dimeric polypeptide chain on ribosomes, which has no biological activity, that is, the inactive precursor protein (pro-TGF-*β*). After entering the Golgi apparatus, pro-TGF-*β* is further processed, modified, and secreted from cells. Finally, the pro-TGF-*β* is hydrolyzed by a protease or non-protease to form mature TGF-*β* with biological activity. Generally, pro-TGF-*β* releases bioactive TGF-*β* under acidic conditions or after being cleaved by matrix metalloproteinase-2 (MMP-2) and matrix metalloproteinase-9 (MMP-9). Then, it recruits mesenchymal stem cells (MSCs) to the site of bone reconstruction to promote the differentiation and proliferation of osteoblasts and new bone formation. The recruitment of MSCs by TGF-*β* is mainly achieved through the ALK5-Smad2/3-Smad4 signaling pathway. Smad3 plays a major role in this process, while Smad2 plays a minor role [13].

There are three main subtypes of TGF-*β* molecules in mammals, with similar structures but different genetic codes: TGF-*β*1, TGF-*β*2, and TGF-*β*3. The potential TGF-*β*1 is secreted in almost all cells. TGF-*β*2 and TGF-*β*3 are mainly found in epithelial cells and mesenchymal cells, respectively. TGF-*β*1 can not only promote the proliferation of osteoblasts but also induce the differentiation and maturation of osteoblasts [14]. It can transform type I receptors of osteoblasts into phosphorylated functional complexes; TGF-*β*2 binding with TPRI and TPRII on BMSCs’ membrane forms a complex of TGF-*β*2/TPRI/TPRII to activate TPRI and then accelerate the formation of osteoblasts through the Smad protein [15]. TGF-*β*3 can interact with the receptors of TGF-*β*3 on the osteoblast membrane and form phosphorylated functional complexes, which promote the phosphorylated Smad2 and Smad3 proteins to enter the nucleus and regulate the synthesis of transcription factors Runx2 to affect the cellular transcription [16].

TGF-*β* usually exists as a dimer linked by disulfide bonds. Compared with other growth factors, TGF-*β* mildly regulates cell proliferation and differentiation and depends on the dose of TGF-*β*, cell cycle, and cell type. Recent studies showed that low concentrations of TGF-*β* enhanced the proliferation and differentiation of osteoblasts, while high concentrations of TGF-*β* suppressed the proliferation and differentiation of osteoblasts [17,18,19]. How to control the dose of TGF-*β* to maximize its side effects needs to be further studied. The effect of TGF-*β* on osteoblasts is time-dependent and plays different roles at different stages of cell differentiation. TGF-*β* can promote the proliferation, early differentiation, and matrix production of the osteoblast progenitor cells. However, in the later stage, Smad3 inhibits the expression and function of Runx2, and thereby TGF-*β* inhibits the differentiation of osteoblasts and matrix synthesis [20].

In bone tissue, TGF-*β* is mainly synthesized by osteoblasts, osteoclasts, and chondrocytes through autocrine and paracrine pathways [21]. It can induce osteoblasts to synthesize the extracellular matrix and promote the overexpression of the extracellular matrix by osteoblasts. Therefore, TGF-*β* plays an important role in regulating the function of osteoblasts [22]. It not only promotes the division, proliferation, differentiation, and maturation of osteoblasts but also increases their number and activity. TGF-*β* can also induce the differentiation of MSCs into osteoblasts by activating extracellular signal-related kinases, such as c-Jun N-terminal kinase and mitogen-activated protein kinase-P38 [23,24]. In addition, TGF-*β* can inhibit the differentiation and formation of osteoclasts, thereby inhibiting bone resorption [25]. Quinn et al. found that osteoclasts could activate TGF-*β*, stimulate osteoblasts to down-regulate the RANKL expression, and increase the OPG expression to inhibit the osteoclast differentiation of BMSCs [26,27]. An increasing number of studies have shown that the expression of TGF-*β*1 mRNA and ALP activity increased obviously after the addition of an osteogenic inducer to MSCs, which significantly promoted bone formation [28,29].

TGF-*β* participates in the whole process of cartilage occurrence and development [30]. In the early stage of chondrogenesis, TGF-*β* promotes chondrocyte differentiation and inhibits chondrocyte hypertrophy differentiation to ensure the amount of extracellular matrix in articular cartilage. In the late stage of chondrogenesis, the expression of TGF-*β* decreases, and the synthesis of some degradation enzyme inhibitors is induced to maintain the chondrocyte phenotype to ensure the stability and function of articular cartilage, which plays an important role in maintaining the integrity of articular cartilage. In addition, TGF-*β* can effectively regulate the processes of bone resorption and bone remodeling and also has a great regulatory effect on osteoblast replication and bone matrix synthesis [31,32]. TGF-*β* is an important factor in the process of bone formation and absorption, which can stimulate the synthesis of collagen by osteoblasts and osteoblasts, increase the synthesis of cartilage-specific type I collagen and fibronectin, and reduce the synthesis of type II and type X collagen. It can also promote the proliferation and differentiation of mesenchymal cells and osteoblasts. By regulating the expression of proteins, TGF-*β* promotes the secretion of fibronectin and collagen by fibroblasts and accelerates the formation of bone and cartilage.

### 2.2. Bone Morphogenetic Protein (BMP)

Urist found out in 1965 that the bone matrix contained active proteins that promoted the formation of cartilage and bone tissue [33]. This active protein could directionally differentiate undifferentiated mesenchymal cells into osteoblasts, which then formed bone tissue. Therefore, it was named bone morphogenetic protein (BMP) [34]. The structure of the BMP was not identified until 1988 [35]. BMP belongs to the TGF-*β* superfamily; more than 30 family members have been found; the molecular weight is about 30 kDa [36]. They are divided into five subgroups according to the homology of the amino acid series. It should be noted that BMP-1 is not a member of the TGF-*β* family. It not only lacks the conserved region structure characteristic of the TGF-*β* family but is also a zinc-dependent metalloproteinase, thus it is a completely different protein [37]. The BMP monomer consists of a signal peptide, a functional region, and a mature peptide, and each monomer contains a sequence consisting of seven cysteine residues. The precursor of BMP shows high conservation due to the seven cysteine residues in the carboxyl-terminal, which are involved in the connection of two monomers to form the dimer proprotein and finally form a mature protein in the presence of convertase. Both homodimers and heterodimers are produced and secreted to functional sites by disulfide bond polymerization after enzymatic hydrolysis guided by the above cysteine residues [38,39]. Studies have found that heterodimers and homodimers of BMP can interact with surface receptors in osteogenic potential and undifferentiated MSCs, thereby promoting MSCs differentiation, aggregation into cartilage and bone, and finally the formation of bone marrow [40].

Wnt and BMP have their ligands, receptors, transduction proteins, and molecular mechanisms, but they interact and influence each other in the process of regulating cell differentiation and proliferation [41,42]. There are at least four modes of action between Wnt and BMP signaling pathways [43]. The first way is that they play the roles through their independent pathways. For example, the Wnt signaling pathway functions by inhibiting cell differentiation and promoting cell proliferation, while the BMP signaling pathway functions by promoting cell differentiation. The second way is that they are involved in succession at different stages of the organism’s development, each performing its duties and maintaining a state of equilibrium. The third way is that they act together on the same target gene, thus showing additive and synergistic effects. The last way is that the BMP signaling promotes the expression of the Wnt gene, which is often seen in embryonic development and nerve formation. Wei et al. found that the BMP-2/Smad signaling pathway can enhance the activity of *β*-catenin, accelerate the phosphorylation of *β*-catenin, and increase the concentration of *β*-catenin [44]. It is involved in the up-regulation of the LEF-1 molecule downstream of the Wnt/*β*-catenin pathway. Mechanism studies showed that the BMP-2/Smad signaling pathway may affect the activity of the *β*-catenin intracellular regulatory complex GSK-3*β*-APC-Axin through Smurf2. This prevents the degradation of *β*-catenin, thereby enhancing the activity of the Wnt/*β*-catenin pathway [45].

Wnt and BMP proteins, two important signaling molecules, participate in the regulation of bone regeneration, the promotion of stem cell proliferation, and osteogenic differentiation through synergistic effects [46,47]. Li et al. conducted a comprehensive and systematic study on BMP proteins, and the results showed that BMP-2, 4, 6, 7, and 9 could induce MSCs to differentiate into osteoblasts. Among these molecules, BMP-9 had the strongest inductive effect [48]. The canonical Wnt signaling pathway takes part in the osteogenic differentiation of MSCs induced by BMP-9. Tang et al. found that Wnt3a and BMP-9 could mutually increase ALP activity induced by MSCs, whereas FRZ, a Wnt signaling antagonist, could inhibit ALP activity induced by BMP-9 [49]. Immunochromatin co-precipitation analysis showed that *β*-catenin played an important role. In vivo experiments confirmed that BMP-9-induced ectopic osteogenesis and calcium salt deposition could be inhibited by FRZ overexpression or *β*-catenin knockdown.

BMP-2, 4, 5, 6, 7, and 14 are proteins closely related to chondrogenesis and development [50,51]. BMP-2 is one of the most important growth factors for inducing allogenic osteogenesis in vitro and participating in the osteogenic and chondrogenic differentiation of various stem cells [52,53]. The mice with BMP-2 knockouts had disordered chondrocyte growth plate regions, resulting in a delayed primary ossification center and severe defects in chondrocyte proliferation, differentiation, and apoptosis [54]. Through real-time quantitative PCR and Western blot, Claus et al. found that exogenous BMP-2 could induce the expression of chondrocytes and type II collagen and improve the expansion function of chondrocytes at both ends of the knee joint [55]. In addition, BMP-4 can also induce MSCs to differentiate into chondrocytes and promote the maturation of chondrocytes [56]. Steinert et al. found that BMP-4 could induce mouse embryonic stem cells and hMSCs to differentiate into chondrocytes and promote the expression of genes related to chondrocyte hypertrophy [57]. Mechanism studies have shown that *β*-catenin signaling partially stimulates chondrocyte differentiation through the BMP-2/4 signaling pathway and BMP-2 promotes/inhibits the expression of Runx2 in a dose-dependent manner, thereby affecting protein ubiquitination in chondrocytes.

Both Wnt and BMP signaling pathways participate in tooth development [58]. They mutually promote or inhibit tooth development by regulating tooth morphology, number, and structure [59]. Zhang et al. found that Wnt/*β*-catenin and BMP-9 signaling pathways are involved in the osteogenic differentiation of apical dental papilla stem cells [60]. BMP-9 can increase the activity of early ALP and late calcium deposition in immortalized stem cells from the apical papilla (ISCAP) in a concentration-dependent manner. Silencing *β*-catenin can down-regulate the effect of BMP-9 on early and late ISCAP osteogenesis. Although BMP-9 has strong potential to become a new method to enhance bone reconstruction in the clinic, its mechanism has not yet been fully understood. Therefore, further research is needed to ensure that BMP-9 can be safely and effectively applied in clinics.

BMP is important for bone growth because it is an acidic glycoprotein with strong osteogenic properties. This growth factor with multifunctional differentiation participates in the processes of bone formation, differentiation, and absorption and has different degrees of influence on the differentiation and proliferation of osteoblasts, osteoclasts, and chondrocytes [61]. It plays an important role in embryonic development, bone growth, cartilage repair, and regeneration [62]. It is also the key to the osteogenic differentiation of MSCs [63]. In 2002, rhBMP-2 (recombinant human bone morphogenetic protein 2) and rhBMP-7 (recombinant human bone morphogenetic protein 7) were approved by the FDA to treat anterior lumbar fusion and long bone defects, respectively [64]. However, the complex process and high cost of the recombinant proteins limit their clinical application. Therefore, how to provide efficient, cheap, and convenient bone regeneration material technology has become a common concern, and is also an urgent clinical requirement.

### 2.3. Fibroblast Growth Factor (FGF)

The FGF family consists of twenty-three members with molecular weights ranging from 17 kDa to 34 kDa (in vertebrates) [65]. They are polypeptides composed of about 150–200 amino acids and contain 20–50% of the same amino acid sequence. The FGF family is divided into three categories according to its functional characteristics [66]. The first category is the classical FGF, namely paracrine fibroblast growth factors, including FGF1–10, 16–18, 20, and 22, which have good affinity to heparin and heparan sulfate and exert their functions by interacting with cell surface tyrosine kinases. The second category is endocrine fibroblast growth factors [67], including FGF-15/19 (FGF-15 is the mouse homolog of FGF-19), 21 and 23, which must first interact with Klotho before binding to FGFR in target cells and then exert their effects on the downstream molecules. The third category is intracellular fibroblast growth factors, including FGF-11, 12, 13, and 14. These FGFs perform their function intracellularly and do not bind to fibroblast growth factor receptors (FGFRs), acting only as non-signaling proteins for voltage-gated sodium channels and other cofactors.

The fibroblast growth factor (FGF) is widely found in the human bone matrix. As a potential mitogen, it can accelerate cartilage repair in vivo and promote the proliferation, differentiation, and maturation of chondrocytes in vitro [68,69]. The basic fibroblast growth factor (BFGF) is released from necrotic tissue when the body suffers from a fracture or tissue injury [70]. It participates in the self-repair process of bone tissue and promotes osteoblast proliferation and collagen formation. FGF-2, 3, 4, 8, 9, 11, 17, 18, and 23 have been found not only to be involved in bone growth and development but also to participate in the differentiation, proliferation, and migration of osteoblasts, osteoclasts, and chondrocytes [71].

FGF-2 expression in chondrocytes and osteoblasts affects the maturation and development of primary cartilage and growth plate cartilage and cartilage regeneration and significantly stimulates the proliferation and osteogenic ability of BMSCs at the early stage of differentiation [72,73,74]. The expression of FGF-2 in osteoblasts and stromal cells promotes the recruitment of bone marrow stromal cells [75,76]. Some studies showed that FGF-2 promoted bone formation and stimulated bone regeneration [77,78]. However, another study showed that FGF-2 reduced bone formation and even completely inhibited ossification [79]. Knockdown of FGF-2 resulted in an obvious differentiation of osteoblasts and an enhanced mineralized phenotype [80]. There are still different views on the FGF-2 effect on bone metabolism, and the effect on osteoclasts has not attracted enough attention [81].

FGF-23 is a new member of the FGF family with a molecular weight of 32 kDa. It is mainly secreted by bone cells and regulates calcium and phosphorus metabolism by acting on the kidney [82]. FGF-23 participates in osteogenic differentiation through binding and activating tyrosine kinase receptors on the cell surface. The effect of FGF-23 on osteogenic differentiation is related to the concentration of FGF-23 [83]. When the concentration of FGF is too high or too low, it will affect normal bone development and bone metabolism [84]. After transfection with FGF-23 in mice, the mineral density of teeth and the mandible decreased [85]. The volume of mineralized teeth decreased, as did the volume of mineralized mandibular cortical bone and alveolar bone. The tooth volume, restorative dentin area, the expression of dentin sialoprotein, and the deposition of type I collagen and osteocalcin in the dentin matrix were significantly reduced. Lv et al. found that FGF-23 acted on different stages of osteoblast growth, reduced ALP enzyme activity through FGFR-3, inhibited the differentiation potential of BMSCs and phosphate production and absorption, and reduced the generation of mineralized nodules, thus inhibiting osteoblast mineralization [86]. In addition, FGF-23 also plays a non-phosphate-dependent role in osteocytes [87]. In the presence of α-Klotho, FGF-23 inhibits chondrocyte proliferation in vitro and in vivo. FGF-23 knockout mice have significantly lower body weight, femur and tibia length, bone mineral density, and trabecular bone volume [88].

FGF-11 is a factor driving pathological bone resorption in osteolytic diseases, which can be induced by FGF-11 to stimulate bone resorption under hypoxic conditions [89]. It is involved in the odontogenesis process, and FGF-11 mRNA is expressed in epithelial signaling centers at key stages of tooth formation in mice [90]. FGF-18 has a good affinity for FGFR-3 and has a significant inhibitory effect on cartilage hypertrophy-related factors and precursor inflammatory cytokines after binding to FGFR-3 [91]. This significantly enhances anabolic metabolism during articular cartilage repair [73]. It is currently the only FGF-based clinical drug for the treatment of osteoarthritis [92].

The human FGFR family includes four members: FGFR-1, FGFR-2, FGFR-3, and FGFR-4 [93]. Although all four FGFRs are encoded by different genes, their amino acid sequences share 56–71% similarity. Hypertrophic chondrocytes mainly express FGFR-1 and FGFR-2 and participate in intramembrane osteogenesis [94,95]. Osteoblasts in the ossification center mainly express FGFR-2 [96] and a small amount of FGFR-1 and FGFR-3 [97]. Mutations of these receptors will induce chondrocyte dysfunction, cause skull dysplasia, and lead to craniosynostosis. FGFR-3 has the highest affinity for FGF-1 and FGF-2. It has an inhibitory effect on endochondral bone formation, thereby inhibiting long bone growth. Additionally, mutations in the FGFR-3 gene can cause achondroplasia [98].

FGFR is an important signal molecule regulating the development of bone gaps. Its mutation will lead to the development of bone gaps, including achondroplasia, premature closure syndrome of the nape suture, and abnormal metabolism syndrome [99,100,101]. It also participates in the repair process of bone gap injuries and is closely related to the repair of endochondral osteogenesis and intramembrane osteogenesis. In addition, FGFR participates in the repair of bone injuries by regulating the function of osteoclasts [102]. A specific knockout of FGFR-1 in osteoclasts can cause increased bone mass, osteoclast differentiation, and bone resorption dysfunction in mice [103]. FGF-2 activates the MAPK signaling pathway through FGFR-1 binding to osteoclasts and promotes the bone resorption function of osteoclasts [104]. Su et al. found that the loss of FGFR-3 function in osteoclasts did not affect the development of bone pathways in mice, but the mass of long bones increased. After selective FGFR-3 knockout, osteoclast differentiation and morphology were not abnormal, but bone resorption function was reduced and bone healing was slowed [105]. Therefore, FGFR plays a regulatory role in bone development and the proliferation and differentiation of osteoblasts and osteoclasts. FGFR treatments provide new ideas to treat osteoporosis [106].

## 3. Wnt Signaling Pathway

Until now, three Wnt signaling pathways have been found, including the Wnt/*β*-catenin pathway, the non-Wnt/Ca^2+^ pathway, and the Wnt planar cell polarity (PCP) pathway [107]. An increasing number of studies show that the Wnt signaling pathway is related to bone formation and bone remodeling [108,109]. It is highly conserved and widely present in invertebrates and vertebrates. It participates in early embryo development, organogenesis, and tissue regeneration. It regulates the morphology and function of cells and the growth and development of bones. It has been found that the formation and progression of many types of diseases, such as osteoporosis, osteogenesis imperfecta, and tumor, are related to the Wnt signaling pathway [110]. Recent studies indicate that the Wnt gene is associated with the growth, differentiation, and apoptosis of MSCs, which affects bone formation and development and regulates signal transduction in osteoporosis [111].

### 3.1. Wnt/β-Catenin Signaling Pathway

(1)Wnt proteins

The Wnt gene was discovered by Nusse when studying mouse breast cancer and was named lnt-1 at that time [112]. Further experiments showed that this gene has high homology with the wingless gene of Drosophila during embryonic development, so the two genes are collectively called Wnt [113]. Wnt consists of 350–400 amino acids and is a secreted signaling glycoprotein with a molecular weight of 39 kDa–46 kDa. The structure of Wnt proteins was first identified by Janda from the complex of XWnt8/Fz8-CRD (Xenopus Wnt8 and Frizzled-8 cysteine-rich domain) [114]. As shown in Figure 2A–C, the XWnt8/Fz8-CRD complex is like a donut, and Wnt8 is like a crab with two pincers embracing and grasping the frizzled receptor. XWnt8 contains an N-terminal α-helical domain (NTD) and a C-terminal cysteine-rich domain (CTD). XWnt8 and Fz8-CRD bind together through two sites and form a large hole in the center of the complex. The identification of Wnt spatial structure is of great significance for the in-depth understanding of its related signaling pathways and the research of regenerative medicine drugs. Recently, Zhong et al. obtained the three-dimensional structure of the complex of human Wntless (WLS) and Wnt3a with a resolution of 2.2 Å by using single-particle cryoelectron microscopy [115]. The structure shows that three main binding regions are formed between WLS and Wnt3a (Figure 2D). Through biochemical and functional experiments, the interactions within these binding regions are critical for the binding of WLS and Wnt3a, the secretion of Wnt3a, and the activation of the Wnt3a downstream signaling pathway. The amino acid sequence of the interaction region is highly conserved among different Wnt and WLS (Figure 2E,F), suggesting that WLS-mediated Wnt secretion has a conserved molecular mechanism.

To date, more than nineteen Wnt proteins have been discovered (Wnt1, Wnt2, Wnt2b, Wnt3, Wnt4, Wnt5a, Wnt5b, Wnt6, Wnt7a, Wnt8a, Wnt8b, Wnt9a, Wnt9b, Wnt10a, Wnt10b, Wnt11, and Wnt16) [116], which are mainly divided into two types. The first type is canonical Wnt proteins. They bind with FRZ/LRP to play a synergistic role in activating signaling pathways, such as Wnt2, Wnt3, Wnt3a, Wnt8, Wnt8b, and Wnt10b. The other is non-canonical Wnt proteins that combine with FRZ to activate heterotrimeric G-protein and improve intracellular calcium levels, such as Wnt4, Wnt5a, Wnt5b, Wnt6, Wnt7a, and Wnt11. All the Wnt proteins must bind to lipids to activate the signaling pathways. The Wnt lipidation is performed by the modification of porcupine in the endoplasmic reticulum and Wntless (WLS) in the Golgi complex. The lipid moiety enhances the hydrophobicity of the Wnt protein and confines the Wnt protein to the cell membrane or its homologous receptor. The transmembrane protein WLS can only specifically bind with the lipidated Wnt protein and secrete Wnt proteins after transporting them to the plasma membrane [117]. After Wnt secretion, its lipid region plays an important role in limiting Wnt diffusion.

Wnt signaling is a key molecule that determines the directional differentiation of MSCs into osteoblasts and participates in bone formation, development, and remodeling [118,119]. It induces bone marrow progenitor cells to differentiate into osteoblasts by inhibiting the expression of adipocyte transcription factor and osteoblast transcription factor alone or simultaneously. Wnt3a, Wnt5a, Wnt6, Wnt10a, Wnt10b, and Wnt16 promote osteogenic differentiation of MSCs through the Wnt/*β*-cateninn pathway [120,121,122,123]. Zhang et al. found that Wnt3a could also promote the early osteoblast differentiation of ISCAP through the Wnt/*β*-catenin pathway [124]. The mechanism studies showed that Wnt3a promoted the formation and deposition of bone matrix by activating Runx2 through synergistic action with BMP-9 [125]. Silencing the *β*-catenin signaling blocks the entry of *β*-catenin into the nucleus to activate TCF-1, thus failing to activate Runx2. Therefore, when stem cells from the apical papilla (SCAP) are infected with Wnt3a and BMP-9, the osteogenic ability of SCAP cells is significantly enhanced. However, the synergistic mechanism between Wnt3a and BMP-9 needs to be further studied. At the late stage of osteoblast development, the expression of Wnt3b and *β*-catenin is increased, the activity of RANKL is inhibited, and both the number and activity of osteoclasts are decreased.

Wnt5a is essential for maintaining the osteogenic differentiation of human MSCs and acts through mediating Lim-only protein FHL2 [126,127,128]. It is expressed in a distal-to-proximal gradient of growing limb mesenchyme and is associated with developing skeletal elements in hypertrophic prechondrocytes and periosteum [129]. Church found that Wnt5a promoted mesenchymal cells to differentiate into cartilage and inhibited the fat maturation of chondrocytes by differentially regulating the expression of cyclinD1 [130]. Wnt5a can activate not only the Wnt/*β*-catenin pathway but also the Wnt/Ca^2+^ pathway. Which pathway is activated depends on the receptor type on the cell surface. When Wnt5a binds to the Ror2 receptor, it inhibits *β*-catenin transcriptional activity and subsequently blocks the canonical pathway [131]. When Wnt5a binds to the Frizzled and LRP5/6 receptors, it promotes canonical pathway activation [132]. In addition, overexpression of Wnt6, Wnt10a, and Wnt10b can increase the relative expression of *β*-cetenin, suppress 3T-L1 differentiation, and induce osteogenic differentiation [133]. 

(2)*β*-catenin

*β*-catenin is located on human chromosome 3p22-p21.3 and was first discovered by Walt, a German biologist. It is a multifunctional protein with a molecular weight of 88 kDa and is composed of 781 amino acids. It contains one carbon-terminal domain, one nitrogen-terminal domain, and twelve armadillo (ARM) repeats. The carboxy-terminal domain consists of 100 amino acids and contains a catenin binding site as well as GSK-3*β* and CKI phosphorylation sites, which regulate the transcription of downstream target genes. ARM repeats are conserved regions in *β*-catenin. They are composed of 36 α-helices enclosed by rod-like superhelices, which can bind to E-cadherin, Axin, and APC proteins. These proteins can not only improve the stability of the *β*-catenin structure but also coordinate the interaction between *β*-catenin and GSK-3*β* [134]. The amino-terminal domain consists of 130 amino acids and is rich in Ser/Thr residues, which control the stability of *β*-catenin. *β*-catenin is the core molecule of the canonical Wnt signaling pathway, which is responsible for transmitting upstream and downstream molecular signaling.

As an adhesive molecule, *β*-catenin is mainly distributed in the cell membrane or in the form of a complex in the cytoplasm. A small amount of free *β*-catenin molecules can be easily degraded by proteases in the cytoplasm [135]. The protein complexes formed by *β*-catenin and E-cadherin have been shown to participate in signal transduction between different cells. In the absence of the Wnt ligand, *β*-catenin mostly binds to E-cadherin in the cell membrane and makes it attach to the cytoskeletal protein actin, which mediates cell-cell adhesion. When the expression of *β*-catenin was up-regulated, the adhesion between cells was enhanced, and the invasion and metastasis of cells were not easy to occur in the cancer environment. On the contrary, when the expression of *β*-catenin is down-regulated, the adhesion between cells is destroyed, and the cells are more susceptible to invasion and metastasis. For a small amount of *β*-catenin, Axin in the APC/Axin/GSK-3 complex forces GSK-3*β* to move closer to *β*-catenin and together with APC promotes the phosphorylation of *β*-catenin, which is finally degraded through the ubiquitin-proteasome pathway. At this time, TCF/LEF binds with different transcriptional repressor proteins and inhibits the expression of Wnt target genes [136]. The concentration of *β*-catenin is related to cell type and organ type. Particularly, in cancer cells, the change in *β*-catenin concentration is more obvious.

Generally speaking, *β*-catenin has a direct effect on the osteoblastic precursor cells and osteoblasts [137]. As an inducible signaling molecule, *β*-catenin may indirectly induce MSCs to differentiate into osteoblasts by enhancing their response to BMP-2 [138]. However, some studies have shown that *β*-catenin stimulates the activation of the OPG gene promoter in osteoblasts, thereby promoting osteoblast differentiation [139]. In addition, *β*-catenin is related to the regulation of cartilage formation and differentiation not only in the early embryonic stage but also in postnatal cartilage and bone formation. The concentration of *β*-catenin mRNA and protein increases obviously in OA patients and mechanically injured human articular cartilage tissues [140]. Transgenic mice with increased expression of *β*-catenin in articular chondrocytes show decreased articular cartilage tissue, cleft formation, fibrosis, and osteophytes [141]. When *β*-catenin is knocked down, cartilage enlargement and maturation are inhibited [142]. The mechanism studies show that *β*-catenin may act by activating the Ihh and Runx2 signaling or inhibiting PTHrP and IGF-1 signaling [143,144].

(3)Mechanisms of Wnt/*β* signal transduction

In the Wnt/*β*-catenin pathway, *β*-catenin is the core component. As shown in Figure 3, most of the *β*-catenin in the cytoplasm binds to E-cadherin in the cell membrane and makes it attach to the cytoskeletal protein actin to mediate cell-cell adhesion. A small amount of *β*-catenin forms destruction complexes with Axin, APC, CKI, and GSK-3*β*. Axin in the complex makes GSK-3*β* approach *β*-catenin, and together with APC, accelerates the phosphorylation of *β*-catenin, then combines with beta-transduction repeats-containing E3 ubiquitin ligase protein (*β*-TRCP), and finally is degraded by the ubiquitin-proteasome pathway, so that *β*-catenin in the cytoplasm remains at a low level. Meanwhile, TCF/LEF binds to a variety of transcriptional repressor proteins, such as Groucho family members, to prevent the expression of Wnt target genes.

Dickkopf1 (Dkk-1), secreted frizzled-related protein (SFRP), sclerostin (SOST), and Wnt inhibitor factor 1 (Wif1) are all secreted Wnt inhibitor proteins that can inhibit the activation of the canonical Wnt signaling pathway [145]. Dkk-1 and SOST combine with the low-density lipoprotein receptor-related protein 5/6 (LRP5/6) on the surface of osteoblasts, blocking the Wnt signaling pathway, inhibiting the proliferation and differentiation of osteoblasts, and reducing the expression of osteoprotegerin, thus enhancing the biological effect of nuclear factor κB receptor activator ligand on inducing osteoclast differentiation. SFRP and Wif1 can directly interact with the Wnt ligand, inhibit the binding of the Wnt ligand to Frizzled, and block signal transduction [146]. Inhibition of SFRP1 and Wif1 can activate Wnt signaling pathways. At present, there are few studies on the effects of SFRP and Wif1 on osteoporosis, and there are no related clinical research reports. Therefore, whether they have effects on osteoporosis needs to be further confirmed.

In the presence of a Wnt molecule, the Wnt/*β*-catenin signaling pathway is activated (Figure 4). The Wnt ligand binds to Frizzled (FRZ) and interacts with the co-receptor LRP5/6 to activate tyrosine kinase (CKI) in cells and recruit Dishevelled (DVL) proteins to the cell membrane for phosphorylation. On the one hand, the phosphorylated DVL inhibits the degradation of the phosphorylation of *β*-catenin. On the other hand, it recruits Axin to destroy the Axin-APC-*β*-catenin complex so that *β*-catenin is released from this complex and transported into the nucleus. The *β*-catenin molecule replaces the transcription inhibitor Groucho to bind with T-cell factor/lymphoid enhancing factor (TCF/LET) in the nucleus and convene the transcription activator, thus initiating the transcription of the target gene downstream of the Wnt pathway and expressing cell cycle proteins, such as cyclin D1, Runx2, Osx and MMP. The Wnt signaling pathway regulates gene transcription through the combination of *β*-catenin and TCF/LET family, the key factor of which is to accumulate enough *β*-catenin in the cytoplasm. When the *β*-catenin level in the cytoplasm is low, the Wnt signaling pathway is turned off. When the level is high, the Wnt signaling pathway is turned on.

### 3.2. Wnt/Ca^2+^ Pathway

Proteins involved in the Wnt/Ca^2+^ pathway are Wnt ligand, FRZ receptor, G-protein, DVL protein, PLC, IP3, DAG, PKC, CaM, CaMK Ⅱ, and other components. It participates in regulating the concentration of intracellular Ca^2+^. In the presence of Wnt signaling, the Wnt ligand binds FRZ receptors on the cell membrane to recruit DSH. Subsequently, the trimeric G-protein Gα/G*β*/Gγ binding with DSH activates PLC, which acts on phosphatidylinositol-4,5-diphosphate in the cell membrane to produce inositol triphosphate (IP3) and diacylglycerol (DAG). IP3 activates Ca^2+^ after entering the endoplasmic reticulum, which can promote the increase of intracellular Ca^2+^ concentration, accelerate the formation of the Ca^2+^/CaM complex, and thus activate downstream calcineurin (CaN). Activated CaN dephosphorylates NFAT in cells, then migrates to the nucleus to act on downstream target molecules. DAG activates protein kinase C (PKC) and promotes CaMKⅡ to exert biological effects through autophosphorylation. In general, the Wnt/Ca^2+^ pathway, based on the increase of intracellular Ca^2+^ concentration, activates PKC, CaMK Ⅱ, NFAT, and other signaling and transcription factors and plays a role in regulating cell adhesion and gene expression [147]. 

The Wnt/Ca^2+^ signaling pathway is mainly activated by Wnt5a and Wnt11 [148]. Wnt5a is the only member protein in the Wnt family that is expressed in the bone growth aggregation area and the progression area of limb development. It can promote the generation of osteoclasts. It is also expressed in the epithelium and mesenchymal of the jaw process, jawbone, mandible, Myer’s cartilage, skull base cartilage, and salivary gland to varying degrees. It plays a major role in regulating the differentiation of chondrocytes and osteoblasts [149]. When Wnt5a binds to frizzled receptors, the levels of IP3, deacylcynopicrin (DAC), and Ca^2+^ temporarily increase, which triggers the activation of NF-κB and NFAT to regulate the generation of osteoclasts. The Wnt5a/Frizzled-2/Ca^2+^ pathway has an inhibitory effect on the Wnt/*β*-catenin signaling pathway. In the downstream of the Wnt5a/Frizzled-2/Ca^2+^ pathway, PKC can activate GTPase Cdc42, while CAMK Ⅱ can phosphorylate TGF-*β*-activated kinase 1 (TAK1). It induces the activation of Nemo-like kinase (NLK) and inhibits the transcription of the Wnt/*β*-catenin signaling pathway. Wnt11 participates in the differentiation of MSCs. The stable expression of Wnt11 can increase the concentration of intracellular free calcium ions, thereby promoting the activity of PKC and inhibiting NF-κB activity. Wang et al. found that SPA (S.aureus protein A) could inhibit the non-canonical Wnt/Ca^2+^ signaling pathway mediated by Wnt11 and significantly downregulate the ability of hMSCs to differentiate into osteoblasts in vitro [148]. Wnt11 promotes osteogenic differentiation of hMSCs under conditions of infection and inflammation in vitro. Therefore, it can be used as a potential regulatory target to enhance osteogenic differentiation.

### 3.3. Wnt/Polarity Pathway (PCP)

The Wnt/PCP pathway is a dishevelled (DSH)-dependent pathway mediated by c-jun N-terminal kinase (JNK). It has two signal transduction pathways. The first pathway is that the Wnt protein binds with the FRZ receptor, the Ror2 receptor, or the Ryk receptor to promote DSH to recruit disheveled-associated activator of morphogenesis 1 (DAMM1) and mediates Rho to active Rho-associated protein kinase (Rock), thereby causing cell migration, cytoskeleton, and tissue structure changes. Another pathway is that DSH directly cooperates with Rac1 and JNK to activate nuclear factor c-Jun and activate transcription factor 2 (ATF2), causing transcription, translation, and expression of downstream target genes, thereby regulating cell differentiation and proliferation [149].

The Wnt/PCP pathway is mainly activated by Wnt4, Wnt5a, and Wnt11, of which Wnt5a is most closely related to bone formation. Wnt5a regulates the change of the cytoskeleton by regulating the activity of Rock [150], thus inducing human adipose mesenchymal stem cells (hASCs) to differentiate into osteoblasts. Under mechanical stimulation, Wnt5a promotes osteogenic differentiation of BMSCs by activating the JNK-mediated signaling pathway [151]. In addition, Wnt5a can bind with Ror2 to activate JNK, and then c-Jun binds with the promoter SP1 to promote the RANK expression of osteoclast precursors and bone formation [152].

## 4. Wnt Signaling Pathway and Bone-Related Cells

### 4.1. Wnt Signaling Pathway and BMSCs

The Wnt signaling pathway is closely related to BMSCs, osteoblasts, osteoclasts, and chondrocytes [27]. BMSCs are undifferentiated mesoderm-like cells found in mammalian bone marrow stroma that may differentiate into different cells such as osteoblasts, fat cells, chondrocytes, neurons, adipocytes, and myoblast cells under certain conditions. Wnt signaling is a key molecule that determines the directional differentiation of BMSCs into osteoblasts [47]. Wnt6, Wnt10a, and Wnt10b promote osteogenic differentiation of BMSCs and inhibit adipocyte differentiation through the Wnt/*β*-catenin signaling pathway. Overexpression of Wnt10a increases the relative expression of *β*-catenin and then induces osteogenic differentiation. 

The osteogenic differentiation of BMSCs is regulated by biological molecules such as proteins, RNAs, and cytokines through the Wnt/*β*-catenin pathway. Hang et al. studied the effect of the Apelin protein on the osteogenesis of hBMSCs. The results showed that the expression of *β*-catenin was up-regulated and the mineral deposition increased in the Apelin-treated cells. Inhibitors that specifically inhibit the Wnt/*β*-catenin signaling pathway also decreased Apelin-induced osteogenic differentiation. Therefore, Apelin partially regulates the osteogenic differentiation of BMSCs and effectively promotes fracture healing through the Wnt/*β*-catenin signaling pathway [153]. 

### 4.2. Wnt Signaling Pathway and Osteoblasts

The canonical Wnt signaling pathway plays a crucial role in the differentiation, proliferation, and maturation of osteoblasts and is deeply involved in regulating bone development and bone remodeling. Kim et al. found that albiflorin plays a positive role in osteoblast differentiation in MC3T3-E1 cells. It enhances the expression of bone formation-related genes such as OSN, OCN, and OPN through the Wnt/*β*-catenin signaling pathway [154]. During glucocorticoid-induced osteoblast differentiation, Wnt signaling can promote the differentiation of osteoblasts at the early stage, but its ability is strongly inhibited at the terminal differentiation stage [155]. Intravenous parathyroid hormone (PTH) significantly increased *β*-catenin expression and TCF/LEF transcriptional activity in osteoblasts [156]. Guhong injection (GHI) can also promote osteoblast proliferation, increase the expression of Wnt3a, *β*-catenin, and LEF-1, and accelerate fracture healing, which indicates that the Wnt/*β*-catenin signaling pathway plays positive roles in regulating osteogenic differentiation in vivo and in vitro [157]. 

The expression of *β*-catenin is closely related to osteoblast activity in type I diabetic animals. Moderate activation of *β*-catenin signaling in osteoblasts can significantly enhance bone formation ability in type 1 diabetic mice and effectively improve bone loss and bone strength damage [158]. In addition, Wnt3a stimulates early osteogenic differentiation of ISCAPs, and the dosage of Wnt3a is positively correlated with ALP activity. Wnt3a and BMP-9 play a synergistic role in promoting osteogenic differentiation in ISCAPs. Silencing the *β*-catenin gene can block the translocation of *β*-catenin into the nucleus and activate TCF-1, thereby failing to activate Ru. Therefore, when the ISCAPs are infected with Wnt3a or BMP-9 alone or in combination, the osteogenic differentiation ability of ISCAPs is significantly stronger than that of ISCAP cells with *β*-catenin silencing [159]. Wnt signaling can promote the proliferation and differentiation of osteoblasts by stimulating the canonical Wnt pathway, and also activate the non-canonical Wnt pathway to induce the expression of downstream signals and inhibit the expression of osteoclasts [160]. The embryonic stem cells (ESCs) isolated from Wnt5a knockout mice show defective osteogenic differentiation during osteogenesis in vitro. The key factors of the non-canonical Wnt signaling pathway, such as PKC, CamKII, and JNK, are significantly down-regulated when Wnt5a is overexpressed, indicating that the non-canonical Wnt signaling pathway plays a regulatory role in osteogenic differentiation [161].

The Wnt signaling pathway enables mesenchymal stem cells to conduct downward along the osteoblast differentiation pathway. Sclerostin (SOST) is an important signaling molecule produced in the Wnt/*β*-catenin pathway. The concentration of SOST in osteoblasts and osteocytes is negatively related to bone mass and bone strength. In animal models of osteoporosis, inhibition of SOST by the monoclonal antibodies omosozumab and blosozumab induced osteoblast differentiation and new bone formation, normalized bone mass, and improved bone structure and strength. Clinical trials have also shown that anti-sclerostin antibody treatment can significantly increase bone formation and reduce bone resorption [162]. Parathyroid hormone (PTH) can inhibit the production of SOST by osteocytes, enhance Wnt/*β*-catenin signaling and significantly improve bone mass. Therefore, PTH may serve as a potential drug for the treatment of postmenopausal osteoporosis [163]. In osteoblasts, SOST binding with LRP5/6 inhibits Wnt signaling in the presence of LRP4. Therefore, blocking the interaction between LRP5/6 and SOST is proposed as an effective treatment for osteoporosis. In the Dkk family, Dkk-1 and Dkk-4 significantly inhibit the Wnt signaling pathway; Dkk-2 inhibits or promotes the Wnt signaling pathway; and Dkk-3 did not affect the Wnt signaling pathway [164]. Overexpression of Dkk-1 leads to a decrease in *β*-catenin levels, inhibits osteoblast differentiation, and promotes osteoblast apoptosis. Dkk-1 binds to the Wnt receptor LRP5/6 on the surface of osteoblasts, blocking the Wnt signaling pathway and preventing osteoblast proliferation and differentiation. Qiang et al. found that the number of osteoblasts decreased by 49% and the concentration of serum osteocalcin decreased by 45% in transgenic mice with Dkk-1 overexpression, resulting in serious bone loss [165]. On the contrary, the number of osteoblasts and bone formation rate of heterozygous mice with Dkk-1 deficiency were significantly higher than those of wild-type mice. In addition, Glantschnig et al. effectively treated osteoporosis caused by estrogen deficiency with an anti-Dkk-1 antibody. After the antibody treatment, the total type I procollagen N-terminal peptide in mice increased significantly, and new bone was obviously formed [166]. 

### 4.3. Wnt Signaling Pathway and Osteoclasts

The relevant evidence of Wnt regulation of osteoclasts is mainly indirect [167], meaning that it affects the differentiation and activity of osteoclasts through the regulation of RANKL and macrophage colony-stimulating factor (M-CSF) in osteoblasts [168]. Wnt signaling is enhanced to promote the expression of osteoprotegerin in osteoblasts, thereby reducing osteoclast formation and bone resorption and inhibiting osteoclast differentiation. On the contrary, reduced Wnt signaling induces osteoclast differentiation and promotes bone resorption. Yang et al. showed that BMSCs participated in the migration and differentiation of osteoclasts through Wnt5a/Ror2 signaling, which led to a decrease in bone mass and finally a loss of subchondral trabecular bone [169]. 

## 5. Osteoporosis and Wnt Signaling Pathway

Osteoporosis is a common bone disease that is mainly manifested as decreased bone mass, bone mineral density, bone strength, and increased risk of fracture. Wnt plays an irreplaceable role in bone development, osteoblast differentiation and maturation, and maintenance of normal bone homeostasis. The activation of Wnt signaling can lead to dual effects on the treatment of bone diseases [170]. Knockdown of *β*-catenin significantly increases the number of osteoclasts, enhances bone resorption, and sharply reduces bone mass, which causes osteoporosis. On the contrary, overexpression of *β*-catenin causes a significant increase in the number of osteoblasts and bone mass and enhances bone formation. Therefore, the activation of the Wnt/*β*-catenin signaling pathway can provide a new approach to treating osteoporosis [171]. 

The bone resorption inhibitors include bisphosphonates, estrogen calcitonin, selective estrogen receptor modulators, and Wnt inhibitors. Dickkopf-1 (Dkk-1), an important inhibitor of the Wnt signaling pathway, is involved in bone development and bone remodeling. It inhibits Wnt signaling by competitively binding to the *β*-helix domain of LRP5/6, thereby regulating the transcription of downstream target genes. Heiland found that inhibition of Dkk-1 had an inhibitory effect on inflammatory bone loss in TNF transgenic mice, accompanied by an increase in the expression level of *β*-catenin [172]. This means that inhibition of osteogenesis accelerates the process of inflammatory osteoporosis and suggests that inhibition of Dkk-1 expression may help cure osteoporosis patients. Piters et al. found a strong association between Dkk1 gene variants and osteoporosis in a study of 783 white young men [173]. In addition, patients with disuse bone loss caused by long-term bed rest have increased expression of serum Dkk-1 level and decreased expression of *β*-catenin, accompanied by the decreased bone formation and increased bone resorption [174]. These studies show that the increase of Dkk-1 can reduce the activity of osteoblasts and bone formation. Therefore, inhibiting the expression of Dkk-1 in bone tissue is an effective treatment for osteoporosis.

Anti-Dkk-1 antibodies can effectively alleviate osteoporosis symptoms caused by estrogen deficiency. Glantschnig et al. studied the therapeutic effect of monoclonal anti-Dkk-1 antibodies (RH2-18) on osteoporosis mice caused by estrogen deficiency [175]. After 8 weeks of RH2-18 treatment, the expression of Dkk-1 was effectively inhibited. The bone mineral density of the femur and lumbar vertebrae in osteoporosis mice increased to the same level as that of the sham group, and the trabecular bone microstructure was completely restored. Voskaridou et al. studied the effect of Zoledronic acid on 66 patients with thalassemia-related osteoporosis [176]. After Zoledronic acid treatment, Dkk-1 concentration in serum was obviously decreased, and bone mineral density was remarkably increased in osteoporosis patients. This indicates that neutralizing the Dkk1 antibody or enhancing Wnt/*β*-catenin signaling could effectively improve osteoporosis caused by thalassemia. At present, the research on the Dkk-1 antibody is mainly limited to animal experiments. How to specifically regulate the expression of Dkk-l in bone tissue without affecting the metabolism of other tissues and organs is an urgent problem to be solved.

## 6. Conclusions and Future Directions

The dynamic balance of bone metabolism plays a significant role in maintaining the function of normal bone tissue. When the balance is broken, bone metabolic diseases such as osteoporosis will occur. The Wnt signaling pathway participates in bone development and metabolism and is closely associated with the differentiation and proliferation of MSCs, osteoblasts, osteoclasts, and chondrocytes. Based on the role of the Wnt signaling pathway in bone metabolism, an increasing number of researchers pay attention to the pathogenesis of osteoporosis and develop drugs for the treatment of osteoporosis by inhibiting or activating the Wnt signaling pathway. SOST and Dkk-1 can competitively bind to the co-receptor LRP5/6 in the Wnt signaling pathway and then regulate the transcription of downstream target genes, ultimately leading to the occurrence and development of osteoporosis. Therefore, SOST and Dkk-1 have become very potent targets for the treatment of osteoporosis. Inhibiting the expression of SOST and Dkk-1 and activating the Wnt signaling pathway can effectively increase bone formation and inhibit bone resorption, which is considered an effective method to treat osteoporosis. Compared with traditional anti-resorption drugs, anti-SOST antibodies, and anti-Dkk-1 antibodies can effectively increase bone formation and reverse the decline of bone mineral density in osteoporosis patients. However, the specific mechanism is still unclear due to the complexity of the Wnt signaling pathway. The clinical use of antagonists that inhibit signaling pathways is limited. In addition, the process of bone metabolism is not only regulated by the Wnt pathway but also involves the Notch pathway, Hedgehog pathway, and OPG/RANKL/RANK pathway, and each signaling pathway also affects the other. Therefore, in-depth studies of the mechanism of the Wnt pathway in bone metabolism and its relationship with other signaling pathways will provide new methods to prevent and treat osteoporosis.

## Figures and Tables

**Figure 1 biomolecules-13-00483-f001:**
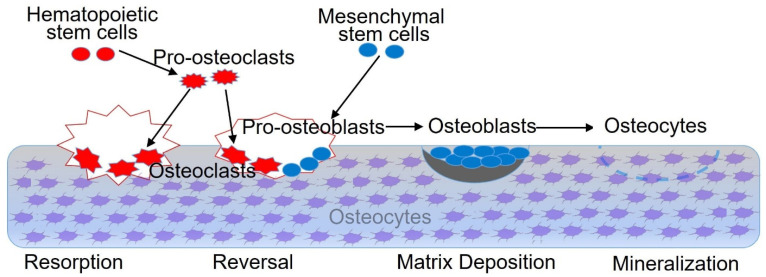
The process of bone remodeling. The bone remodeling process is a dynamic balance. The bone-related cells such as BMSCs, pro-osteoclasts, pro-osteoblasts, osteoclasts, osteoblasts, and osteocytes participate in bone formation and development in the four phases listed above. The osteoblasts are mainly responsible for building and shaping the skeleton and finally differentiating into osteocytes. The osteoclasts are multinucleated cells and can efficiently absorb surface areas of cancellous bone. Osteocytes are non-dividing cells embedded in the bone with several protrusions called canaliculi and have a rather regulatory phenotype.

**Figure 2 biomolecules-13-00483-f002:**
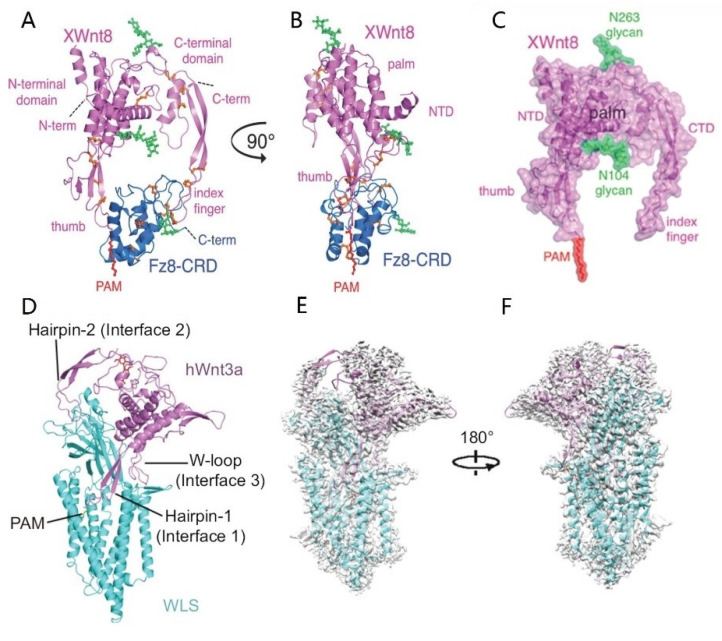
Overall structures of the XWnt8/Fz8-CRD complex (**A**–**C**) [114] and the Wnt3a/WLS complex (**D**–**F**) [115]. The structure is shown in a cartoon with XWnt8 and Wnt3a colored in violet, Fz8-CRD colored in blue, and WLS colored in cyan. (**A**) The spatial structure of XWnt8/Fz8-CRD complex as viewed ‘face on’. (**B**) The spatial structure of the XWnt8/Fz8-CRD complex as viewed ‘side-on’. (**C**) The structure of XWnt8 after moving Fz8-CRD from the complex. (**D**) The spatial structure of the tWnt3a/WLS complex. (**E**) Fit of the WLS/Wnt3a complex model with the cryo-EM map. (**F**) The other side of the fitting panel E after 180 degrees rotation.

**Figure 3 biomolecules-13-00483-f003:**
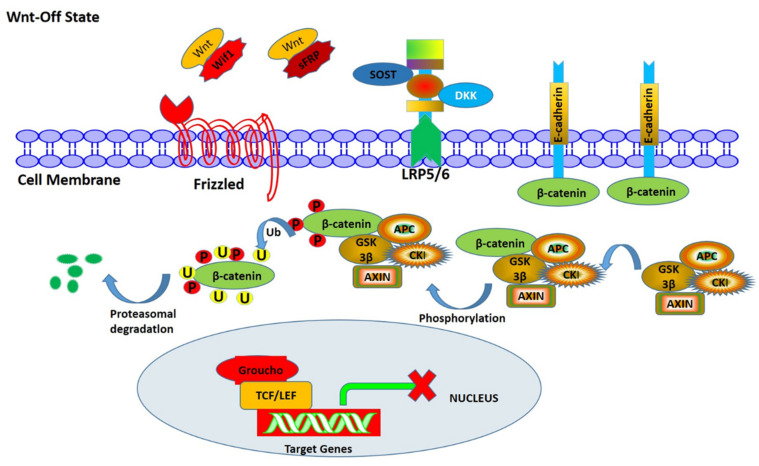
The process of Wnt signal transduction on the inactive state. Without the Wnt ligand, intracellular phosphorylation of *β*-catenin is activated by kinases CKI and GSK-3*β* under the action of the skeleton proteins Axin and APC. After phosphorylation, *β* -catenin combines with *β*-TRCP, which leads to its ubiquitination and degradation. Due to the low level of *β*-catenin in the nucleus, the transcription inhibitor Groucho can prevent the transcription of the Wnt target gene.

**Figure 4 biomolecules-13-00483-f004:**
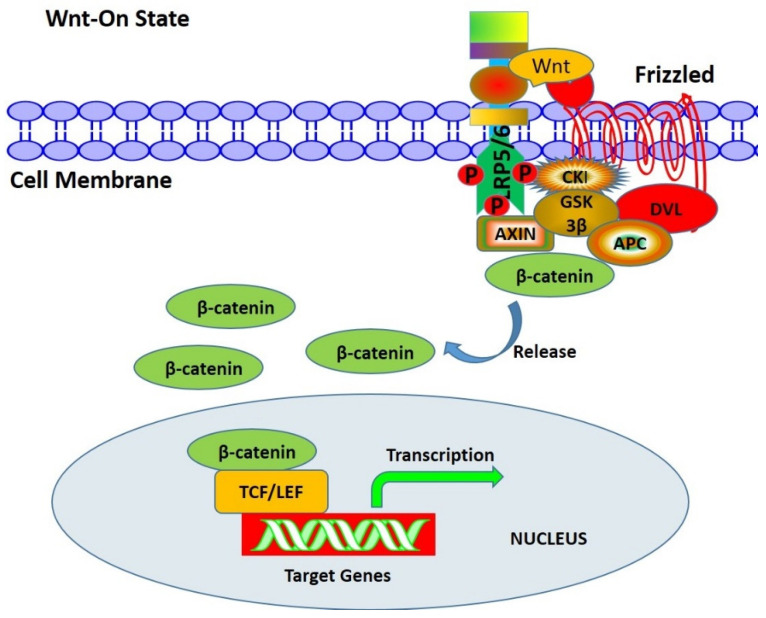
The process of Wnt signal transduction on the active state. In the presence of a Wnt signal, *the* Wnt ligand binds with Frizzled receptor and LRP5/6 co-receptor on the surface of the cell membrane, inducing phosphorylation of DVL through CKI, thus inhibiting GSK-3*β* activity, and *β*-catenin will accumulate in the nucleus. Thus, LEF/TCF-mediated transcription of target genes is activated.

## Data Availability

Not applicable.

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
