# Peer review of "Progress of Wnt Signaling Pathway in Osteoporosis"

_biomolecules, 2023, doi:10.3390/biom13030483_

Round 1
Reviewer 1 Report
An extensive review, very well documented, the result of a long work. Of course, it does not bring major news, but it represents a corollary to what we know about osteoporosis. As the conclusions point out, many more studies will be needed until an ideal treatment of osteoporosis.
Author Response
Many thanks to the reviewer for the comments on this paper. As stated by the reviewer, osteoporosis seriously affects the quality of life of middle-aged and elderly people, but there is no effective method to cure the disease, and more treatment research on osteoporosis is needed. The purpose of this article is to summarize the previous work, so as to attract more people's attention to osteoporosis.
Reviewer 2 Report
In the following, I will comment on the review “Progress of Wnt signaling pathway in osteoporosis” by Yongguang Gao and coworkers. Overall, the study is well designed. However, the authors should check their manuscript for correct literature reference because in some paragraphs the literature coverage is lacking. Also the authors should think about the TGF-β/FGF/BMP chapters. They are too detailed for a wnt review and to superficial to stand on their own. In addition, I would like to have them better integrated in the overall topic of the review.
· Figure 1 Osteocytes are not on the cell surface, inducing mineralization. This task is still done by mature osteoblasts. Osteocytes are non-dividing cells imbedded into the bone with several protrusions called canaliculi and have a rather regulatory phenotype. These cells express SOST which is anti-mineralization.
· L95 is not clear what do you mean?
· L97 if you really want to address the activation of TGF-β you should do it properly. Please look up the additionally to the latency associated protein (LAP), which is the propeptide, the latent tgf-β binding protein up.
· L100 MMPs are not stimulating, they are the active partner. Thus, they are either cleaving or activating as enzymes.
· L129 Proteins are not secreted in an autocrine or paracrine way. They induce signaling in an autocrine or paracrine way. And are you sure that osteoclasts secret TGF-β at all?
· If you really want to include TGF-β you should address the collagen regulation by TGF-β because it plays a rather important role in bone generation and osteoporosis.
· BMP-1 is not only lacking the structure of TGF-β it is a zinc dependent metalloproteinase, thus it is a completely different protein.
· L164 is a strange sentence
· 174-l191 this paragraph let me think about the order of data. Here you expect knowledge of the Wnt pathway, which you should have described in the wnt chapter. After all this is a wnt review. Therefore I would suggest a paragraph in the ending which addresses the wnt signaling in context with the other presented pathway. In that case all the members are properly established when you discuss the interactions between them.
· L231 Why are BMPs acidic?
· L250 affinity to
· L333 conserved
· Figure 2 You should put the source of the figures if you did not made themselves in the subtitles of the figure.
· L372 a table would be nice to keep track about the wnts its main signaling pathway and the implication in bone/osteoporosis.
· L420 +426 usually it is called amino-terminal and carboxy-terminal
· L436 are you describing a cancer environment, because you mention metastases.
· L447 molecule
· Figure 3 / Figure 4– The style could be more pleasant. With the grass green background it is difficult to notice the β-cathenin, which is the most important protein on this sketch.
· L505 who binds?
· L510 Rather “Proteins involved in the Wnt/Ca2+ pathway are ….”
· L525-537 for the whole paragraph all literature references are absent. There are several statements concerning WNT5a without any proof.
· L565-573 again literature is missing
· L619 which receptor is blocked in particular by dickkopf?
· Chapter 4.3 This is a bit superficial. Either you explain the function of chondrocytes, the role of transdifferentiation into osteoblasts, fibrosis and phenotypic stability in more detail. Or you omit the chondrocyte aspect completely
· L675 as wnt inhibitor you should start with SOST in the dickkopf chapter since the inhibition mechanism is similar
· L687-701 There are many redundancies between this chapter and the paragraph starting in L613. Please make a general inhibitor part and an wnt inhibitor part in osteoporosis
·
Altogether, I recommend this review fit for publication after major revision.
Author Response
1) Figure 1 Osteocytes are not on the cell surface, inducing mineralization. This task is still done by mature osteoblasts. Osteocytes are non-dividing cells imbedded into the bone with several protrusions called canaliculi and have a rather regulatory phenotype. These cells express SOST which is anti-mineralization.
Re: Thank you very much for your reminding. We are very sorry for the misunderstanding caused by this part of the content. We have revised Figure 1 according to your comments and made a text description.
2) L95 is not clear what do you mean?
Re: Sorry for the unclear expression. There are more than 40 members in TGF-β superfamily: TGF-β, activin, inhibin, bone morphogenetic protein (BMP) and so on. Above expression has been added in the text.
3) L97 if you really want to address the activation of TGF-β you should do it properly. Please look up the additionally to the latency associated protein (LAP), which is the propeptide, the latent tgf-β binding protein up.
Re: The newly secreted TGF-β gene (per-pro-TGF-β) combines with latency associated protein (LAP) and forms a dimeric polypeptide chain on ribosomes, which has no biological activity, that is, the inactive precursor protein (pro-TGF-β). After entering the Golgi apparatus, pro-TGF-β is further processed, modified and secreted from cells. Finally, the pro-TGF-β is hydrolyzed by protease or non-protease to form mature TGF‐β with biological activity.
4) L100 MMPs are not stimulating, they are the active partner. Thus, they are either cleaving or activating as enzymes.
Re: We agree with the reviewer. The pro-TGF-β releases bioactive TGF‐β under acidic conditions or after being cleaved by matrix metalloproteinases. The right expression has been added in the text.
5) L129 Proteins are not secreted in an autocrine or paracrine way. They induce signaling in an autocrine or paracrine way. And are you sure that osteoclasts secret TGF-β at all?
Re: Thank you very much for your reminding. In bone tissue, TGF-β is mainly synthesized by osteoblasts, osteoclasts and chondrocytes through autocrine and paracrine pathways. It can induce osteoblasts to synthesize extracellular matrix and promote the over-expression of extracellular matrix of osteoblasts. Above expression has been added in the text.
6) If you really want to include TGF-β you should address the collagen regulation by TGF-β because it plays a rather important role in bone generation and osteoporosis.
Re: Thanks for the good suggestion. “collagen regulation by TGF-β” has been added in the text.
7) BMP-1 is not only lacking the structure of TGF-β it is a zinc dependent metalloproteinase, thus it is a completely different protein.
Re: Thanks for the reminder. We have made further statements in the text.
8) L164 is a strange sentence
Re: The sentence has been deleted.
9) 174-191 this paragraph let me think about the order of data. Here you expect knowledge of the Wnt pathway, which you should have described in the wnt chapter. After all this is a wnt review. Therefore I would suggest a paragraph in the ending which addresses the wnt signaling in context with the other presented pathway. In that case all the members are properly established when you discuss the interactions between them.
Re: Thanks for the good suggestion. The process of bone metabolism is not only regulated by the Wnt pathway, but also in-volves the Notch pathway, Hedgehog pathway and OPG/RANKL/RANK pathway and each signaling pathway also affects each other. Therefore, in-depth study of the mechanism of Wnt pathway in bone metabolism and its relationship with other signaling pathways will provide new methods to prevent and treat osteoporosis. Above contents are stated at the end of the text.
10) L231 Why are BMPs acidic? L250 affinity to; L333 conserved
Re: BMP is rich in glutamic acid, and each glutamic acid molecule contains 2 carboxyl groups, so BMP is an acidic glycoprotein. Carboxylic acid has a high affinity with hydroxyapatite, which is also the fundamental reason why BMP can play an important role in bone repair. These two mistakes have been corrected.
11) Figure 2 You should put the source of the figures if you did not made themselves in the subtitles of the figure.
Re: Thanks for the reminder. The source of the figures has been added in the caption.
12) L372 a table would be nice to keep track about the wnts its main signaling pathway and the implication in bone/osteoporosis.
Re: Thanks for the suggestion. Although 19 Wnt proteins have been discovered so far, it is not clear by which signaling pathway some of them participate in life activities, so we have not made a table.
13) L420 +426 usually it is called amino-terminal and carboxy-terminal
Re: They have been corrected.
14) L436 are you describing a cancer environment, because you mention metastases.
Re: Sorry for the unclear expression. “In cancer environment“ has been added.
15) L447 molecule
Re: It has been corrected.
16) Figure 3 / Figure 4– The style could be more pleasant. With the grass green background it is difficult to notice the β-cathenin, which is the most important protein on this sketch.
Re: Thanks for the suggestion. The grass green background has been removed.
17) L505 who binds?
Re: Wnt ligand binds with Frizzled receptor and LRP5/6 co-receptor, which has been revised.
18) L510 Rather “Proteins involved in the Wnt/Ca2+ pathway are ….”
Re: Thanks for the suggestion. Good expression has been used in the text.
19) L525-537 for the whole paragraph all literature references are absent. There are several statements concerning WNT5a without any proof.
Re: The references have been added in the text.
20) L565-573 again literature is missing
Re: The references have been added in the text.
21) L619 which receptor is blocked in particular by dickkopf?
Re: Wnt receptor LRP5/6 is blocked in particular by dickkopf, which has been revised.
22) Chapter 4.3 This is a bit superficial. Either you explain the function of chondrocytes, the role of transdifferentiation into osteoblasts, fibrosis and phenotypic stability in more detail. Or you omit the chondrocyte aspect completely
Re: The chondrocyte aspect has been deleted.
23) L675 as wnt inhibitor you should start with SOST in the dickkopf chapter since the inhibition mechanism is similar
Re: Thanks for the suggestion. This section has been moved to the dickkopf chapter.
24) L687-701 There are many redundancies between this chapter and the paragraph starting in L613. Please make a general inhibitor part and an wnt inhibitor part in osteoporosis
Re: The general inhibitors and wnt inhibitors have been added at the beginning of this paragraph.
Reviewer 3 Report
In the present review, Yongguang Gao and co-workers analyzed the mechanisms of Wnt proteins, β-catenin and signaling molecules in the process of signal transduction and summarized the relationship between the Wnt signaling pathway and bone-related cells. This latter could offer new perspectives and approaches to make a diagnosis and give a treatment for osteoporosis. Overall, I think that the manuscript is well-structured (within the scope of "Biomolecules”) and of clinical impact on a current topic of interest.
I have some small suggestion/curiosity to improve the quality of review.
1) The current literature demonstrates convincing associations between diet, food components and osteoporosis management. Indeed, it has been suggested that the rational intake of isoflavones, particularly abundant in Soybeans and soy foods, could be very useful in the prevention and cure of postmenopausal osteoporosis (i.e., Bitto, A. et al. Curr Med Chem. 2010;17(27):3007-18, Zhenyu, W. and Luying, L. Front Pharmacol 2022 Sep 8; 13:1016981, Harahap, I.A. and Suliburska, J. Foods. 2021 Nov 3;10(11):2685). Then, please to discuss on the possible application of nutraceutics, probiotics and/or food components that, in combination with healthy diet and physical activity, could provide a possible further strategy to prevent/treat osteoporosis.
2) Gut microbiota composition is related to bone homeostasis as well as to pathophysiology of osteoporosis. Please discuss this very intriguing topic of current research (i.e., Zhong X. et al. J Microbiol Biotechnol. 2021 Jun 28;31(6):765-774).
Author Response
1) The current literature demonstrates convincing associations between diet, food components and osteoporosis management. Indeed, it has been suggested that the rational intake of isoflavones, particularly abundant in Soybeans and soy foods, could be very useful in the prevention and cure of postmenopausal osteoporosis (i.e., Bitto, A. et al. Curr Med Chem. 2010;17(27):3007-18, Zhenyu, W. and Luying, L. Front Pharmacol 2022 Sep 8; 13:1016981, Harahap, I.A. and Suliburska, J. Foods. 2021 Nov 3;10(11):2685). Then, please to discuss on the possible application of nutraceutics, probiotics and/or food components that, in combination with healthy diet and physical activity, could provide a possible further strategy to prevent/treat osteoporosis.
Re: We agree with the reviewer that the rational intake of isoflavones, particularly abundant in Soybeans and soy foods, could be very useful in the prevention and cure of postmenopausal osteoporosis. However, the topic of the article is “Wnt signaling pathway in osteoporosis“ We think that diet and food components in the suppression of osteoporosis are not well suited to the topic of this paper.
2) Gut microbiota composition is related to bone homeostasis as well as to pathophysiology of osteoporosis. Please discuss this very intriguing topic of current research (i.e., Zhong X. et al. J Microbiol Biotechnol. 2021 Jun 28;31(6):765-774).
Re: We agree with the reviewer that gut microbiota composition is related to osteoporosis, but it not well suited to the topic of this paper.
Round 2
Reviewer 2 Report
Before publication I like two things to change
1. Do you really think that osteocytes look like that? Even in schematics and cartoons there is an universally agreed form to depict osteocytes. If you copy from me that these cells have protrusions, then you should at least show them.
2.L 100 it is pre-pro-TGF-beta
Otherwise the corrections are acceptable.
Author Response
1. Do you really think that osteocytes look like that? Even in schematics and cartoons there is an universally agreed form to depict osteocytes. If you copy from me that these cells have protrusions, then you should at least show them.
Re: Sorry for the misunderstanding. We have redrawn the osteocytes in FIG. 1
2. L100 it is pre-pro-TGF-beta
Re: It has been corrected.